# Novel Arylsulfonylhydrazones as Breast Anticancer Agents Discovered by Quantitative Structure-Activity Relationships

**DOI:** 10.3390/molecules28052058

**Published:** 2023-02-22

**Authors:** Violina T. Angelova, Teodora Tatarova, Rositsa Mihaylova, Nikolay Vassilev, Boris Petrov, Zvetanka Zhivkova, Irini Doytchinova

**Affiliations:** 1Faculty of Pharmacy, Medical University of Sofia, 1000 Sofia, Bulgaria; 2Laboratory “Nuclear Magnetic Resonance”, Institute of Organic Chemistry with Centre of Phytochemistry, Bulgarian Academy of Sciences, 1113 Sofia, Bulgaria

**Keywords:** sulfonyl hydrazones, MCF-7, MDA-MB-231, QSAR, breast cancer, anticancer activity

## Abstract

Breast cancer (BC) is the second leading cause of cancer death in women, with more than 600,000 deaths annually. Despite the progress that has been made in early diagnosis and treatment of this disease, there is still a significant need for more effective drugs with fewer side effects. In the present study, we derive QSAR models with good predictive ability based on data from the literature and reveal the relationships between the chemical structures of a set of arylsulfonylhydrazones and their anticancer activity on human ER+ breast adenocarcinoma and triple-negative breast (TNBC) adenocarcinoma. Applying the derived knowledge, we design nine novel arylsulfonylhydrazones and screen them in silico for drug likeness. All nine molecules show suitable drug and lead properties. They are synthesized and tested in vitro for anticancer activity on MCF-7 and MDA-MB-231 cell lines. Most of the compounds are more active than predicted and show stronger activity on MCF-7 than on MDA-MB-231. Four of the compounds (**1a**, **1b**, **1c**, and **1e**) show IC_50_ values below 1 μM on MCF-7 and one (**1e**) on MDA-MB-231. The presence of an indole ring bearing 5-Cl, 5-OCH_3_, or 1-COCH_3_ has the most pronounced positive effect on the cytotoxic activity of the arylsulfonylhydrazones designed in the present study.

## 1. Introduction

Breast cancer (BC) is the most common cancer among women worldwide, accounting for 25% of all cancers, and is the second leading cause of cancer death in women after lung cancer [1]. In 2020, an estimated 2.3 million new cases of breast cancer were diagnosed globally, and 627,000 women died from the disease [1]. The most recent data from the American Cancer Society estimate that about one in eight (12%) women in the United States will develop invasive breast cancer at some point in their lives [2].

BC is categorized into three major types based on its molecular characteristics: hormone-based BC (estrogen receptor (ER^+^) or progesterone receptor (PR^+^)), human epidermal receptor 2-expressing (HER2^+^) BC, and triple-negative (ER^−^, PR^−^, and HER2^−^) BC (TNBC) [3]. The type of BC determines the therapeutic approach. The treatment of hormone-based BC involves hormone therapy, which works by inhibiting the production or action of hormones that fuel the growth of cancer cells. Some common types of hormone therapy for BC include tamoxifen (a selective estrogen receptor modulator (SERM) that blocks the effects of estrogen on BC cells) [4] and aromatase inhibitors (a class of drugs that block the production of estrogen by inhibiting the enzyme aromatase) [5]. The CDK4/6 inhibitors (which block the activity of the cyclin-dependent kinases 4 and 6, which play a role in the regulation of the cell cycle) [6], HER2 inhibitors (which block the activity of the HER2 protein, which is overexpressed in some types of BC) [7], and luteinizing hormone-releasing hormone (LHRH) agonists (a class of drugs that lower estrogen levels by inhibiting the production of luteinizing hormone, which is needed for the ovaries to produce estrogen) [8] also have found a place in BC therapy. In addition to hormone therapy, other treatment options for hormone-based BC may include chemotherapy, radiation therapy, surgery, and targeted therapies that block the signaling pathways that promote cancer cell growth [9]. More problematic is the treatment of TNBC, which accounts for 10–15% of BC cases and is characterised by limited possibilities for targeted therapy, as TNBC cells do not overexpress estrogen, progesterone, or HER2/neu receptors. The standard treatment for TNBC typically involves a combination of conventional chemotherapy, surgery, and radiation therapy [10].

Despite the advances in BC therapy, the need for new and more effective drugs with fewer side effects remains. Some promising areas of research in this field include targeted therapies with improved cancer selectivity, that are aimed to specifically target the cancer cells while minimizing harm to normal cells [9], and immunotherapies, which help to boost the body’s own immune system to fight the cancer [11].

Recently, two research groups have independently developed novel arylsulfonylhydrazones as anticancer agents against human BC cells. Senkardes et al. have synthesized and tested a series of sulphonyl hydrazones with anticancer activity on human breast adenocarcinoma cell line MCF-7 and prostate cancer cell line PC-3 [12]. The anticancer activities were in the micromolar range and the selectivity index (*SI* = *IC*_50_ on non-cancer cells/*IC*_50_ on cancer cells) has reached 432 for some of the compounds. Additionally, good cyclooxigenase-2 (COX-2) inhibitory activity has been found in vitro for some of the hydrazones. COX-2 is a proinflammatory enzyme and is overexpressed in solid tumours such as BC and prostate cancer. Gaur et al. have synthesized and tested a series of arylsulfonylhydrazones with indole and morpholine moieties [13]. The compounds have shown anticancer activity on MCF in micromolar concentrations, with a *SI* up to 60. Furthermore, the compounds have been active on the TNBC cell line MDA-MB-468 with *IC*_50_ in the lower micromolar range and with a *SI* up to 37.

In the present study, we analyse the available data for arylsulfonylhydrazones by Quantitative Structure-Activity Relationship (QSAR) modelling. QSAR modelling is a computational technique that has proven to be valuable in the field of anticancer research. QSAR models use mathematical algorithms to analyse and predict the biological activity of chemicals based on their molecular structure. This information can then be used to identify new, promising compounds for further study and development as potential anticancer drugs. Several studies have demonstrated the utility of QSAR in anticancer research by identifying new candidate compounds for specific cancer targets and by facilitating the design of more selective and effective drugs [14,15,16]. In addition, QSAR can provide insights into the molecular mechanisms underlying a compound’s activity, which can help guide the optimization of its structure for improved efficacy and safety [17]. Overall, QSAR modelling represents a powerful tool in the discovery and development of novel anticancer drugs.

We utilize the most effective QSAR models derived in the present study to design a set of potential new anticancer agents. These compounds undergo in silico screening for drug likeness, and the most promising ones are subsequently synthesized and evaluated in vitro on breast cancer cell lines (Figure 1).

## 2. Results

### 2.1. Quantitative Structure–Activity Relationship (QSAR) Models for Arylsulfonylhydrazones as Breast Anticancer Agents

Two sets of arylsulfonylhydrazones were collected from the literature [12,13] and used as a training set for the derivation of QSAR models. The compounds and their anticancer activities, expressed as ligand efficiency (*LE*), are given in Table 1. *LE* measures the ligand activity per non-hydrogen atom and is calculated according to:$$LE=\frac{pIC_{50}}{N}$$
where *pIC*_50_ is the negative decimal logarithm of *IC*_50_ and *N* is the number of non-hydrogen atoms in the molecule. The *LE* values ranged from 0.105 to 0.207 and from 0.110 to 0.170 for the activities on MCF-7 and MDA-MB-468, respectively.

The structures were optimized and described by 70 molecular descriptors, as explained in Materials and Methods. The most relevant descriptors were selected by a genetic algorithm using software tool MDL QSAR v.2.2 (MDL Information Systems Inc., 2004). All possible subset regressions among the selected descriptors were calculated and only models with *r*^2^ ≥ 0.6 and *q*^2^ ≥ 0.4 were considered.

The best model for anticancer activity on cell line MCF-7 is given below:*LE* (MCF-7) = −0.004 × *morph* + 0.015 × *SaaaC_acnt* −0.029 × *SaaN_acnt* −0.012 × *ka*1 + 0.367
where *n =* 26*; r*^2^ = 0.796; *SEE* = 0.014; *q*^2^ = 0.647; *CVRSS* = 0.007; *r*^2^*_random_(mean)* = 0.155, *morph* is a user-defined indicator differentiating the two subsets in the training set; *SaaaC_acnt* accounts for the number of aromatic *aaaC*-atoms in the molecule; *SaaN_acnt* corresponds to the number of aromatic *aaN*-atoms; *ka*1 is first order *kappa alpha* shape index; *r*^2^—goodness of fit, *SEE*—standard error of estimation, *q*^2^—leave-one-out cross validation coefficient; *CVRSS*—cross validation residual sum of squares, and *r*^2^*_random_(mean)*—the mean value of *r*^2^*_random_* values calculated for 100 randomizations of the dependent variable among the compounds. 

The values of the descriptors relevant to the cytotoxic activity on MCF-7 are given in Appendix A. The indicator *morph* takes 1 for the subset **5a**–**k** and 0 for the subset **3a**–**o**. The negative coefficient for *morph* means that the substituent 1-(4-morpholinylethyl)-1H- indol-3-yl in **5a**–**k** is not favourable for *LE* on MCF-7. The descriptor *SaaaC_acnt* varies from 0 (for **3a**–**o**) to 2 (for most of **5a**–**k**) and 4 (for **5h** and **5i**, containing fused rings). Its coefficient in the model is positive, i.e., more *aaaC*-atoms in the molecule correspond to better anticancer activity. The descriptor *SaaN_acnt* takes value 1 for **3h** and **5i**, containing pyrazolyl and quinolyl, respectively. For the rest of the compounds, *SaaN_acnt* takes the value 0. As its coefficient is negative, the presence of aromatic N-atoms of type *aaN* is not essential for the anticancer activity. The kappa shape indices account for the molecular shape [18]. A higher value for *ka*1 corresponds to more branched molecules (more paths). In the training set, the values for *ka*1 vary from 13.666 for **3n** to 25.609 for **5h**. The average *ka*1 for the subset **3a**–**o** is 16.701, for the subset **5a**–**k**—22.421. The negative coefficient for *ka*1 favors the less branched molecules. 

The QSAR model for cytotoxic activity on cell line MDA-MB-468 was derived only on the compounds from the subset **5a**–**k**. The best model is given below:*LE* (MDA-MB-468) = 0.020 × *nelem* − 0.004 × *nvx* + 0.151
where *n =* 11*; r*^2^ = 0.979; *SEE* = 0.003; *q*^2^ = 0.931; *CVRSS* = 0.0002; *r*^2^*_random_(mean)* = 0.155, *nelem* is the number of chemical elements in the molecule and *nvx* accounts for the number of graph vertices. The values of the descriptors relevant to the cytotoxic activity on MDA-MB-468 are given in Appendix A. The number of elements in the molecules **5a**–**k** is five (*C*, *O*, *N*, *S*, and *H*); only **5d** has an additional *F* and **5f** has an additional *Cl*. As the coefficient for *nelem* is positive, obviously, the presence of *F* and *Cl* favours the cytotoxic activity. The range of *nvx* values is from 29 for **5c** to 39 for **5h** and **5i**. The negative coefficient means that the bulky branched substituents are not favourable for the activity on MDA-MB-468. 

The structure–activity relationships found in the derived QSAR models are used next in the design of novel arylsulfonylhydrazones with anticancer activity.

### 2.2. Design of Novel Arylsulfonylhydrazones Based on QSAR Models 

The requirements obtained from the above QSAR models were implemented in the design of novel arylsulfonylhydrazones as anticancer agents, i.e.:For Ar1: Single aromatic ringsFor Ar2: Aromatic rings containing *aaaC* and *Cl* but no *aaN*.

The structures of the designed molecules are given in Table 2. For Ar1, we selected phenyl or 4-methylphenyl substituents. The N-tosyl hydrazones (p-Me-Ph-SO2-NH-N=Ar(R)) are a special class of hydrazones with proven anticancer activity against TNBC cell lines [19]. 

For Ar2, we selected indole and phenyl substituents. The indole ring possesses anti-BC activity [20] due several different signaling pathways [21]. The indole system contains *aaaC* and no *aaN*. The N-atoms in the selected indole substituents are *aaNH* or *daaN*, with slight NH-acidic (*pKa* 16.2) properties. Six of the nine new hydrazones contain mono-substituted indole moiety (compounds **1a**–**e**, **1i**). For comparison, we included three compounds with bi-substituted phenyl rings (compounds **1f**–**h**). Two of the compounds contain the favourable *Cl* atom (compounds **1e** and **1h**). 

The *LE* values of the designed compounds were predicted by the derived models. All of them are close to or higher than the maximum *LE* of the compounds from the training set on both cell lines. At this stage of the study, all designed compounds appeared to be prospective anticancer agents.

### 2.3. In Silico Screening of the Designed Compounds for Drug Likeness

Prior to synthesis, the designed structures were screened in silico for drug likeness considering their physicochemical and ADME properties and pharmacokinetic (*PK*) parameters. 

#### 2.3.1. Physicochemical Properties

The main physicochemical properties calculated for the designed arylsulfonylhydrazones are given in Table 3. They are molecular weight, *Mw; pKa* value; fraction of the ionized molecules, *f_A_; logP*; distribution coefficient at pH 7.4 *logD*_7.4_; polar surface area, *PSA*; count of free rotatable bonds, *FRB*; hydrogen bond donors, *HBD;* hydrogen bond acceptors, *HBA*; count of the violations of Lipinski’s Rule of 5, *R*5.

The molecular weights are around 300 g/mol (295–355 g/mol), which is in a good agreement with the recommended Mw for lead compounds [22]. The compounds are weak acids with *pK_a_* values between 8.59 and 9.09. At pH 7.4, the neutral molecules dominate as indicated by the negligible fraction of ionized molecules *f_A_* and the close values between *logP* and *logD*_7.4_. The *logP* values are around 3, which is, again, in good agreement with the requirements for lead compounds. *PSA*s range from 67 to 92 Å, suggesting good oral absorption and inability to cross the blood–brain barrier (*BBB*) [23]. The number of free rotatable bonds is between 3 and 5; however, the single bonds in the Ar1–S–N–N = fragment are quite rigid due to p-π conjugation. The number of hydrogen bond donors obeys the ‘Rule of 3’; however, the hydrogen bond acceptors exceed it. Regarding Lipinski’s rule of 5, all compounds meet the four criteria and there is no violation. 

#### 2.3.2. ADME Properties

The ADME properties calculated in the study are given in Table 4. The water solubility was calculated by three methods [24] and the average value in mol/L is presented as *logS*. According to the *logS* scale [24], compounds with *logS* between −6 and −4 are considered as moderately soluble, while those with *logS* between −4 and −2—as soluble. According to the BOILED-Egg diagram [23] (Figure 2), all compounds have good oral permeability, one of them (compound **1h**) is able to cross the blood–brain barrier (*BBB*), and none of the compounds are a substrate of the P-glycoprotein (*P-gp*) transporter. The parameter *oral BA* summarizes six criteria which definine the suitable physicochemical space for oral bioavailability [23]. These are lipophilicity (*logP*), size (*Mw*), polarity (*PSA*), solubility (*logS*), insaturation (*fraction of Csp*3 *atoms*), and flexibility (*number of rotatable bonds*). Each criterion has a certain range. The designed compounds violate in insaturation, i.e., the fraction of Csp3 atoms is below the lower limit of 0.25. This violation was expected as most of the C-atoms in the structures are in sp2-hybridization. The *BA score* indicates the probability of bioavailability being higher than 10% in rats [25]. In our case, the probability is 55%. The CYP inhibition considers the five enzymes that most-commonly take part in drug metabolism: 1A2, 2C19, 2C9, 2D6, and 3A4. The studied compounds are able to inhibit between 2 and 4 of the CYPs. Apart from following Lipinski’s rule, all compounds demonstrate *drug likeness* filtered by the criteria of Ghose [26], Veber [27], Egan [28], and Muegge [29]. The *lead likeness* is defined by three criteria: *Mw* in the range 250–350 g/mol, *logP* up to 3.5, and up to 7 rotatable bonds in the molecule [30]. Here, again, our compounds fit well in the ranges. Finally, the synthetic feasibility of the designed compounds was assessed by the synthetic accessibility score, which ranges from 1 (very easy synthesis) to 10 (very difficult synthesis). A score between 2.56 and 2.80 points to relatively easy synthesis. 

#### 2.3.3. Pharmacokinetic Parameters

The main pharmacokinetic parameters, fraction of the unbound-to-plasma-proteins molecules, *fu*; total clearance, *CL*; steady-state volume of distribution, *VDss*; half-life, *t*_1/2_, of the designed arylsulfonylhydrazones were calcuculated by QSPkR models previously derived in our Lab [31,32,33]. The predicted values are given in Table 5.

The *fu* values ranged from 0.010 to 0.074, suggesting high plasma protein binding of all compounds (>90%). It is generally accepted that neutral drugs bind with variable affinity to both human serum albumin and alpha−1-acid glycoprotein [34]. Lipoproteins also contribute to plasma protein binding, especially for highly lipophilic drugs [35].

Total *CL* values ranged between 0.017 and 0.647 L/h/kg. Most of the compounds can be classified as low *CL* drugs, while **1c** and **1d** have medium *CL*. Analysis of a data set of 754 drugs with different ionization states revealed that 78% of anionic and zwitterionic drugs have low *CL* (<0.24 L/h/kg) and only 1–2% have high *CL* (>0.96 L/h/kg). For neutral drugs, these percentages were as follows: 45% low *CL*, 39% moderate *CL*, and 16% high *CL* [36]. Considering the relatively high lipophilicity of the compounds and the negligible ionization at pH 7.4, clearance can be considered to be dominated by metabolism. Neutral drugs have a low renal *CL_R_* unless their *logD*_7.4_ is negative. For drugs with *logD*_7.4_ > 0, the *CL_R_* decreases with lipophilicity due to tubular reabsorption [37].

Values for *VDss* vary between 0.587 and 0.953 L/kg, which is in the order of total body water volume. It is likely that the compounds are evenly distributed throughout the body without significant accumulation in certain tissues and organs.

The half-life (*t*_1/2_) is determined by CL and VDss. Therefore, compounds **1a**, **1c**, **1d**, and **1i**, with medium *CL* and/or low *VDss*, have short *t*_1/2_ (0.94–3.07), while compounds **1b**, **1e**, **1f**, **1g**, and **1h**, with low *CL* and high *VDss*, show moderate to long *t*_1/2_ (6.20–35.12 h).

### 2.4. Synthesis of the Novel Arylsulfonyl Hydrazones

The designed compounds showed strong drug and lead likeness in in silico screening procedures and we decided to synthesize and test all of them.

The arylsulfonylhydrazones were prepared by a condensation reaction (Figure 1) between the corresponding aldehydes and benzenesulfonohydrazide or 4-methylbenzenesulfonohydrazide, at a molar ratio of 1:1, in absolute ethanol for 1–3 h, as described elsewhere [38].

The structures were confirmed by ^1^H NMR, ^13^C NMR, and HRMS spectroscopic data and melting points. The ^1^H-NMR spectra of **1a**–**i** have single signals corresponding to resonances of azomethine protons (CH=N) at 7.82–8.26 ppm. The hydrazide/hydrazone N/H protons are observed at 11.37–11.93 ppm. The ^13^C-NMR spectra exhibit resonances arising from azomethine (C=N) from 130.94 to 147.45, respectively (Appendix A).

### 2.5. Anticancer Activity of the Novel Arylsulfonyl Hydrazones

The anticancer activity of the novel arylsulfonylhydrazones was tested on two BC cell lines: MCF-7 and MDA-MB-231. The cell line MCF-7 originates from human breast adenocarcinoma and expresses estrogen receptor alpha (ER-α) [39], while the cell line MDA-MB-231 represents TNBC adenocarcinoma and lacks any receptor [40]. To test the cytotoxicity of the compounds on healthy cells, they were incubated within Neuro-2a cells, which are mouse neuroblasts isolated from brain tissue [41,42]. The results from the in vitro tests are summarized in Table 6. 

The differences (errors) between the experimental and the predicted *LE* values are given in Table 1. The positive values correspond to underpredicted activity, the negative—to overpredicted activity. The errors range between −0.047 and 0.063 for MCF-7 and from −0.027 to 0.081 for MDA-MB-231. Most of the compounds are more active than expected. Only compounds **1i** and **1g** are less active on MCF-7 and MDA-MB-231, respectively.

The experimental *IC*_50_ values of the novel compounds on MCF-7 range from 0.6 μM to 164.9 μM. The *LE*s are between 0.158 and 0.286, with an average value of 0.230. For comparison, the average *LE* of the training set on the same cell line is 0.156 (0.171 for the subset **3a**–**o** and 0.135 for the subset **5a**–**k**) with the highest value being 0.207. The selectivity index *SI* is defined as the ratio of *IC*_50_ on healthy cells and *IC*_50_ on cancer cells. A *SI* higher than 10 is considered to belong to a selective compound [43]. The *SI*s of the novel compounds span from 0.747 to 46 on MCF-7. Four of the nine compounds show cytotoxic activities on MCF-7 below 1 μM. These are compounds **1e**, **1a**, **1b**, and **1c**. The most efficient compounds on MCF-7 are **1e** and **1a**, while the most selective are compounds **1d** and **1c**.

The most active, efficient, and selective compound on MDA-MB-231 is **1e**, with an *IC*_50_ of 0.9 μM, *LE* of 0.275, and *SI* of 7.222. Compounds **1a**, **1b**, and **1c** have *IC*_50_s in the lower micromolar range with *LE*s around and above 0.2; however, they have low *SI*s. 

## 3. Discussion

Based on data from the literature, QSAR models were obtained in the present study to reveal the relationship between the structures of arylsulfonylhydrazones and their anticancer activity against BC. It was found that, for the activity against ER+ BC, measured on a MCF-7 cell line, a less-branched aromatic substituent with more *aaaC*-atoms, *Cl*, and no *aaN*-atoms performed better as anticancer agents. Less-branched aromatic moieties bearing *F* and *Cl* are required for activity against TNBC, as measured in the MDA-MB-231 cell line. These findings were implemented in the design of nine arylsulfonyl hydrazones. The structures contain mono- and/or bi-substituted phenyl and indolyl moieties. *Cl* atoms were included in two of them. The anticancer activities on both cell lines, expressed as *LE*, were predicted by the derived QSAR models. All compounds demonstrated higher than or close to the maximal *LE*s of the compounds from the training set. Prior to synthesis, the structures were screened in silico for drug likeness by calculating their physicochemical and ADME properties and main PK parameters, such as fraction of the unbound to plasma protein molecules, *fu*; total clearance, *CL*; steady-state volume of distribution, *VDss;* and half-life, *t*_1/2_. In terms of drug likeness, all nine of the designed compounds were suitable as leads. They were synthesized and tested. The in vitro tests confirmed the predicted activities. What is more, seven and eight of the compounds are more active on MCF-7 and MDA-MB-231, respectively, than predicted. Most of the designed compounds are more active on MCF-7 than on MDA-MB-231. The *IC*_50_ values for **1e**, **1a**, **1b**, and **1c** on MCF-7 are below 1 μM. On MDA-MB-231, only compound **1e** shows activity below 1 μM. 

The most active and most efficient compound on both cell lines is **1e**, with a *SI* of 13 for MCF-7 and 7 for MDA-MB-231. It contains a phenyl ring as an Ar1 substituent and 5-chloroindole as an Ar2 subsitituent. Further, **1e** obeys drug and lead likeness rules, has high GI absorption, and has no *BBB* permeability. In terms of *PK* behavior, **1e** is predicted to be extensively bound to plasma proteins (only 1% free fraction), with a total clearance of 5 L/h and a *VDss* of 54 L for a 70-kg patient, as well as a half-life of 7 h. 

The next-most active and efficient arylsulfonylhydrazone on both cell lines is **1a**, with a *SI* of about 9 for MCF-7 and only 1.7 for MDA-MB-231. Further, **1a** bears phenyl as Ar1 and 5-methoxyindole as Ar2. This compound is predicted to be a good drug candidate and lead compound in terms of physicochemical and ADME properties, with extensive plasma–protein binding, a total clearance of 13.5 L/h, a *VDss* of 41 L, and a half-life of 2 h.

Next in activity and efficiency on MCF-7 line are compounds **1b**, **1c**, **1d**, and **1f**. Compounds **1c** and **1d** demonstrate the highest selectivity of 40 and 46, respectively, followed by **1f** with a SI of 18. Compounds **1g**, **1h**, and **1i** are less active, efficient, and selective.

For MDA-MB-231, compounds **1b** and **1c** show activities in the low micromolar range and efficiencies around 0.2; however, they show poor selectivities (up to 2). The remaining compounds are less active and non-selective.

The analysis of substituents shows that the indole ring has the most pronounced positive effect on the cytotoxic activity of the arylsulfonylhydrazones designed in the present study. The substitution of indole by phenyl dramatically reduces the activity on both cell lines (from 10-fold to more than 300-fold on MCF-7 and from 70-fold to complete loss of activity on MDA-MB-231). Among the substituents on the indole ring, 5*-Cl*, 5*-OCH*_3_, and 1*-COCH*_3_ increase the activity between 92- and 330-fold on MCF-7 compared with the 1*-CH*_3_ substituent. The effects of these substituents on the activity on MDA-MB-231 are moderate. The *Cl* atom deserves special attention. Attached to an indole moiety, it increases activity 330-fold on MCF-7 and 70-fold on MDA-MB-231 compared to when it is attached to the phenyl ring.

In conclusion, the QSAR-guided strategy for the design of novel arylsulfonylhydrazones with anticancer activity, applied in the present study, generated several prospective leads with *IC*_50_ values below 1 μM and *SI* values up to 46. The newly designed compounds were more active than the compounds from the training set and represent a starting point for further lead optimization. 

## 4. Materials and Methods

### 4.1. Materials and Reagents

The reagents for the synthesis were analytical or chemically pure and obtained from Sigma–Aldrich (Steinheim, Germany). The solvents used were of analytical grade. The structures of the new molecules were proven by 1H-NMR, 13CNMR, and HRMS spectral data. Their purity was determined by TCL characteristics and melting points.

The in vitro antineoplastic activity of the newly synthesized compounds was evaluated against human BC cell lines of different molecular types: the triple negative MDA-MB-231 cell line and the ER/PR/Her2 positive variant MCF-7, as well as against mouse neuroblast cells, Neuro-2a. All cell lines were purchased from the German Collection of Microorganisms and Cell Cultures (DSMZ GmbH, Braunschweig, Germany) and cultivated according to supplier’s instructions. Cells were cultured in an RPMI 1640 growth medium supplemented with 10% fetal bovine serum (FBS) and 5% L-glutamine, and incubated under standard conditions of 37 °C and 5% humidified CO_2_ atmosphere. 

### 4.2. QSAR Protocol

The training set for the development of QSAR models consisted of 26 compounds. Fifteen compounds were derivatives of 4-methylphenyl hydrazone [12]. The remaining 11 compounds were morpholinylethylindolyl derivatives [13]. The anticancer activities of both subsets were measured in vitro by MTT tests on MCF-7 cell line. The second set was tested on MDA-MB-468 cell line as well. The chemical structures were modeled and optimized by MM+ force field, steepest descent algorithm, and RMS gradient of 0.1 kcal/A.mol using HyperChem 7.52 (Hypercube Inc., Gainesville, FL, USA, 2005). 

The chemical structures were described by 70 descriptors divided into eight groups: atom-type E-state indices, atom-type E-state accounts, hydrogen E-state categories, internal H-bonds E-state indices, kappa shape indices, molecular properties (*logP*, molecular weight, number of elements, number of rings, number of hydrogen-bond donors and acceptors, etc.), 3D descriptors (dipole, polarizability, surface, volume, etc.), and user-defined (*morph*). The descriptor *morph* accounts for the presence of an indole-morpholine fragment in the molecule. If an indole-morpholine is presented in the molecule, *morph* takes 1, otherwise it takes 0. The relevant descriptors were selected by genetic algorithm (GA) at the following settings: size of initial population 32, tournament selection, uniform crossover, one-point mutation, and Friedman’s lack-of-fit scoring function with parameter 2. All possible subset regressions among the selected descriptors were calculated and only models with *r*^2^ (goodness of fit) ≥ 0.6 and *q*^2^ (leave-one-out cross validation coefficient) ≥ 0.4 were considered. To check the validity of the selected descriptor set, 100 randomizations of the dependent variable among the compounds were carried out and *r*^2^*_random_* values were calculated for each regression. If the mean value of *r*^2^*_random_* was lower than *r*^2^, the selected descriptor set was considered as valid. QSAR models were derived by MDL QSAR v.2.2 (MDL Information Systems Inc., 2004).

### 4.3. In Silico Screening for Drug Likeness

The physicochemical properties of the designed compounds were calculated by ACD/LogD tool v. 9.08 (ACD/Labs, Toronto, Canada). The ADME properties were calculated by SwissADME tool [20]. The PK parameters were calculated by previously derived QSPkR models [31,32,33]. As the fraction of the ionized molecules of most of the designed arylsulfonylhydrazones was below 3%, the predictions were based on the QSPkR models derived for neutral molecules. Separate QSPkR models have been derived for the fraction of neutral molecules unbound to plasma proteins, *fu*; unbound clearance of neutral drugs, *Clu*; and steady state volume of distribution of basic and neutral drugs, *VDss*. The datasets consisted of 117 neutral molecules or 407 basic and neutral drugs, respectively, extracted from Obach’s database—the largest and best curated source of data for the key pharmacokinetic parameters after *iv* administration [44]. The chemical structures of the compounds have been encoded by more than 113 to 138 molecular descriptors calculated by ACD/LogD tool v. 9.08 and MDL QSAR version 2.2. Genetic algorithm and step-wise multiple linear regression have been applied for variable selection and model derivation. The QSPkRs have been evaluated by internal and external validation procedures.

### 4.4. Synthesis 

#### 4.4.1. General Information

The nuclear magnetic resonance (NMR) experiments were carried out on a Bruker Avance spectrometer at 600 MHz at 20 °C in deuterated dimethyl sulfoxide (DMSO-d6) as a solvent, and tetramethylsilane (TMS) as an internal standard. The precise assignment of the ^1^H and ^13^CNMR spectra was accomplished by measurement of two-dimensional (2D) homonuclear correlation (correlation spectroscopy (COSY)), DEPT-135, and 2D inverse detected heteronuclear (C–H) correlations (heteronuclear single-quantum correlation spectroscopy (HMQC) and heteronuclear multiple bond correlation spectroscopy (HMBC)). Mass spectra were measured on a Q Exactive Plus mass spectrometer (ThermoFisher Scientific) equipped with a heated electrospray ionization (HESI-II) probe (Thermo Scientific, Bremen, Germany). The melting points were determined using a Buchi 535 apparatus and melting point meter M5000 apparatus. We used IUPAC nomenclature for naming of the newly synthesized compounds.

#### 4.4.2. General Procedure for the Synthesis of the Compounds **1a**–**i**

The solution of 20 mmol of the corresponding carbonyl compounds in 10 mL of absolute ethanol was mixed with a hot solution of 20 mmol (60 °C) benzenesulfonohydrazide or 4-methylbenzenesulfonohydrazide in 10 mL of absolute ethanol and stirred for 1–3 h. Upon cooling, the obtained crystalline precipitates were filtered, washed with ethanol-ether, recrystallized from ethanol, and dried. The new compounds were colorless, white, and light-yellow crystalline solids, stable at normal conditions and soluble in methanol, acetonitrile, and DMSO; poorly soluble in water and ethanol.

*N’-[(Z)-(*5*-methoxy-*1*H-indol-*3*-yl)methylidene]benzenesulfonohydrazide,* **1a**

Yellow solid. Yield: 90%; m.p. 174–175 °C. ^1^H NMR (600 MHz, DMSO-d_6_) δ 3.74 (s, 3H, CH_3_), 6.79 (dd, *J* = 2.6, 8.8 Hz, 1H, H-6), 7.28 (d, *J* = 8.8 Hz, 1H, H-7), 7.43 (d, *J* = 2.5 Hz, 1H, H-4), 7.60 (tt, *J* = 1.7, 7.1 Hz, 2H, H-m), 7.64 (tt, *J* = 1.8, 7.3 Hz, 1H, H-p), 7.67 (d, *J* = 2.8 Hz, 1H, H-2), 7.93 (td, *J* = 1.6, 6.5 Hz, 2H, H-o), 8.08 (s, 1H, CH=N), 10.94 (s, 1H, NH), 11.40 (d, *J* = 2.0 Hz, 1H, NH-indol). ^13^C NMR (151 MHz, DMSO-d_6_) δ 55.15 (CH_3_), 103.03 (C-4) 110.77 (C-3), 112.55 (C-7), 112.65 (C-6), 124.46 (C-3a), 127.35 (C-o), 129.08 (C-m), 130.95 (C-2), 131.79 (C-7a), 132.85 (C-p), 139.17 (C-i), 145.82 (CH=N), 154.38 (C-5). HREIMS *m*/*z* [M + H]^+^ 330.090688 (calcd for C_16_H_15_N_3_O_3_S, [M + H]^+^ 330.09057).

*N’-[(E)-(*5*-methoxy-*1*H-indol-*3*-yl)methylidene]-*4*-methylbenzenesulfonohydrazide,***1b**

Light-yellow solid. Yield: 86%; m.p. 201–202 °C. ^1^H NMR (DMSO-d_6_) δ(ppm): 2.34 (s, 3H, CH_3_), 3.75 (s, 3H, OCH_3_), 6.79 (dd, *J* = 2.6, 8.8 Hz, 1H, H-6), 7.28 (d, *J* = 8.8 Hz, 1H, H-7), 7.39 (d, *J* = 8.0 Hz, 2H, H-3′ and H-5′), 7.45 (d, *J* = 2.5 Hz, 1H, H-4), 7.65 (d, *J* = 2.8 Hz, 1H, H-2), 7.81 (d, *J* = 8.3 Hz, 2H, H-2′ and H-6′), 8.06 (s, 1H, CH=N), 10.82 (s, 1H, NH), 11.37 (s, 1H, NH-indol). NOESY: between H-2′(H-6′) and NH-N; CH and NH-N; CH and H-2(H-4), which proves the E orientation. ^13^C NMR (DMSO-d_6_) δ(ppm): 20.94 (CH_3_), 55.10 (OCH_3_), 103.07 (C-4), 110.81 (C-3), 112.49 (C-7), 112.62 (C-6), 124.45 (C-3a, s), 127.36 (C-2′ and C-6′), 129.44 (C-3′ and C-5′), 130.79 (C-2), 131.77 (C-7a), 136.29 (C-1′), 143.15 (C-4′), 145.52 (CH=N), 154.35 (C-5). HREIMS *m/z* [M + H]^+^ 344.10547 (calcd for C_17_H_17_N_3_O_3_S, [M + H]+ 344.106338).

*N’-[(E)-(*1*-acetyl-*1*H-indol-*3*-yl)methylidene]benzenesulfonohydrazide,* **1c**

Light-yellow solid. Yield: 89%; m.p. 209–210 °C. ^1^H NMR (DMSO-d_6_) δ(ppm): 2.63 (s, 3H, CH_3_), 7.33–7.40 (m, 2H, H-5 and H-6), 7.59–7.66 (m, 3H, H-3′, H4′ and H-5′), 7.93 (dd, *J* = 1.7, 7.9 Hz, 2H, H-2′ and H-6′), 8.05 (dd, *J* = 1.6, 7.7 Hz, 1H, H-4), 8.10 (s, 1H, H-2), 8.26 (s, 1H, CH=N), 8.32 (dd, *J* = 1.4, 6.7 Hz, 1H, H-7), 11.47 (s, 1H, NH-indol). ^13^C NMR (DMSO-d_6_) δ(ppm): 23.78 (CH_3_), 115.87 (C-4), 116.17 (C-3), 122.00 (C-7), 124.21 (C-5), 125.69 (C-6), 126.36 (C-3a), 127.21 (C-2′ and C-6′), 129.21 (C-3′ and C-5′), 131.08 (CH=N), 133.05 (C-4′), 135.69 (C-7a), 138.90 (C-1′), 142.61 (C-2), 169.56 (C=O). HREIMS *m/z* [M + H]+ 342.08989 (calcd for C_17_H_15_N_3_O_3_S, [M + H]^+^ 342.090688).

*N’-[(E)-(*1*-acetyl-*1*H-indol-*3*-yl)methylidene]-*4*-methylbenzenesulfonohydrazide,* **1d**

Light-yellow solid. Yield: 87%; m.p. 211–212 °C. ^1^H NMR (DMSO-d_6_) δ(ppm): 2.33 (s, 3H, CH3), 2.63 (s, 3H, COCH3), 7.33–7.40 (m, 2H, H-5 and H-6), 7.40 (d, *J* = 8.0 Hz, 2H, H-3′ and H-5′), 7.81 (d, *J* = 8.3 Hz, 2H, H-2′ and H-6′), 8.07 (dd, *J* = 1.7, 5.8 Hz, 1H, H-4), 8.08 (s, 1H, CH=N), 8.25 (s, 1H, H-2), 8.32 (dd, *J* = 1.6, 6.4 Hz, 1H, H-7), 11.39 (bs, 1H, NH-indol). ^13^C NMR (DMSO-d_6_) δ(ppm): 20.96 (CH3), 23.79 (COCH3), 115.87 (C-4), 116.26 (C-3), 122.06 (C-7), 124.20 (C-5), 125.68 (C-6), 126.40 (C-3a), 127.26 (C-2′ and C-6′), 129.61 (C-3′ and C-5′), 130.94 (CH=N), 135.70 (C-7a), 136.08 (C-1′), 142.34 (C-2), 143.42 (C-4′), 169.56 (C=O). HREIMS *m/z* [M + H]^+^ 356.10542 (calcd for C_18_H_17_N_3_O_3_S, [M + H]^+^ 356.106338).

*N’-[(E)-(*5*-chloro-*1*H-indol-*3*-yl)methylidene]benzenesulfonohydrazide,* **1e**

Yellow solid. Yield: 81%; m.p. 183–184 °C. ^1^H NMR (600 MHz, DMSO-d_6_) δ 7.17 (dd, *J* = 2.1, 8.6 Hz, 1H, H-6), 7.41 (d, *J* = 8.6 Hz, 1H, H-7), 7.62 (t, *J* = 7.3 Hz, 2H, H-m), 7.66 (t, *J* = 7.2 Hz, 1H, H-p), 7.80 (d, *J* = 2.7 Hz, 1H, H-2), 7.89 (d, *J* = 2.0 Hz, 1H, H-4), 7.92 (d, *J* = 7.0 Hz, 2H, H-o), 8.08 (s, 1H, CH=N), 11.05 (s, 1H, NH), 11.70 (bs, 1H, NH-indol). ^13^C NMR (151 MHz, DMSO-d_6_) δ 110.68 (C-3), 113.47 (C-7), 120.73 (C-4), 122.56 (C-6), 125.02 (C-5), 125.09 (C-3a), 127.37 (C-o), 129.14 (C-m), 131.94 (C-2), 133.02 (C-p), 135.36 (C-7a), 138.98 (C-i), 144.88 (CH=N). HREIMS *m/z* [M + H]^+^ 334.041151 (calcd for C_15_H_12_ClN_3_O_2_S, [M + H]^+^ 334.04123). 

*N’-[(E)-(*3*,*4*-dimethoxyphenyl)methylidene]benzenesulfonohydrazide,***1f**

White solid. Yield: 87%; m.p. 150–152 °C. ^1^H NMR (600 MHz, DMSO-d_6_) δ 3.76 (s, 3H, OCH3), 3.76 (s, 3H, OCH_3_), 6.95 (d, *J* = 8.4 Hz, 1H, H-5), 7.08 (dd, *J* = 1.9, 8.3 Hz, 1H, H-6), 7.12 (d, *J* = 1.9 Hz, 1H, H-2), 7.61 (tt, *J* = 1.6, 7.5 Hz, 2H, H-m), 7.66 (tt, *J* = 1.7, 11.1 Hz, 1H, H-p), 7.83 (s, 1H, CH=N), 7.88 (td, *J* = 2.1, 7.7 Hz, 2H, H-o), 11.33 (s, 1H, NH). ^13^C NMR (151 MHz, DMSO-d_6_) δ 55.42 (OCH_3_), 55.56 (OCH_3_), 108.58 (C-2), 111.49 (C-5), 121.00 (C-6), 126.36 (s, 1C), 127.26 (C-o), 129.21 (C-m), 133.05 (C-p), 139.01 (C-i), 147.45 (CH=N), 148.90 (C-3), 150.66 (C-4). HREIMS m/z [M + H]+ 321.090353 (calcd for C15H16N2O4S, [M + H]+ 321.0895).

*N’-[(E)-(*3*,*4*-dimethoxyphenyl)methylidene]-*4*-methylbenzenesulfonohydrazide,* **1g**

White solid. Yield: 82%; m.p. 174–175 °C. ^1^H NMR (DMSO-d_6_) δ(ppm): 2.36 (s, 3H, CH_3_), 3.76 (s, 6H, OCH_3_), 6.95 (d, *J* = 8.3 Hz, 1H, H-5), 7.08 (dd, *J* = 1.9, 8.3 Hz, 1H, H-6), 7.12 (d, *J* = 1.8 Hz, 1H, H-2), 7.40 (d, *J* = 8.1 Hz, 2H, H-3′ and H-5′), 7.76 (d, *J* = 8.3 Hz, 2H, H-2′ and H-6′), 7.82 (s, 1H, CH=N), 11.21 (s, 1H, NH). ^13^C NMR (DMSO-d_6_) δ(ppm): 20.98 (CH_3_), 55.41 (OCH_3_), 55.54 (OCH_3_), 108.62 (C-2), 111.50 (C-5), 120.92 (C-6), 126.41 (C-1), 127.27 (C-2′ and C-6′), 129.57 (C-3′ and C5′), 136.14 (C-1′), 143.36 (C-4′), 147.18 (CH=N), 148.89 (C-3), 150.61 (C-4). HREIMS m/z [M + H]+ 335.10518 (calcd for C16H18N2O4S, [M+H]^+^ 335.106003).

*N’-[(E)-(*4*-chlorophenyl)methylidene]benzenesulfonohydrazide,***1h**

White solid. Yield: 80%; m.p. 161–163 °C. ^1^H NMR (600 MHz, DMSO-d_6_) δ 7.45 (td, *J* = 2.2, 9.1 Hz, 2H, H-3 and H-5), 7.58 (td, *J* = 2.2, 9.1 Hz, 2H, H-2 and H-6), 7.61 (tt, *J* = 1.5, 7.2 Hz, 2H, H-m), 7.67 (tt, *J* = 1.6, 7.1 Hz, 1H, H-p), 7.88 (td, *J* = 1.5, 6.6 Hz, 2H, H-o), 7.91 (s, 1H, CH=N), 11.64 (s, 1H, NH). ^13^C NMR (151 MHz, DMSO-d_6_) δ 145.90 (CH=N), 127.18 (C-o), 128.44 (C-2 and C-6), 128.93 (C-3 and C-5), 129.32 (C-m), 132.56 (C-1), 133.16 (C-p), 134.60 (C-4), 138.95 (C-i). HREIMS *m/z* [M + H]^+^ 295.030252 (calcd for C_19_H_17_NO_4_, [M + H]^+^ 295.03044).

*N’-[(E)-(*5*-methoxy-*1*-methyl-*1*H-indol-*3*-yl)methylidene]benzenesulfonohydrazide*, **1i**

Yellow solid. Yield: 83%; m.p. 190–191 °C. ^1^H NMR (600 MHz, DMSO-d_6_) δ 3.73 (s, 3H, NCH_3_), 3.75 (s, 3H, OCH_3_), 6.85 (dd, *J* = 2.6, 8.9 Hz, 1H, H-6), 7.35 (d, *J* = 8.9 Hz, 1H, H-7), 7.63 (tt, *J* = 1.4, 7.4 Hz, 1H, H-p), 7.64 (s, 1H, H-2), 10.93 (s, 1H, NH-indol), 7.44 (d, *J* = 2.5 Hz, 1H, H-4), 7.59 (tt, *J* = 1.7, 7.3 Hz, 2H, H-m), 7.92 (dd, *J* = 1.5, 7.0 Hz, 2H, H-o), 8.05 (s, 1H, CH=N). ^13^C NMR (151 MHz, DMSO-d_6_) δ 32.94 (NCH_3_), 55.21 (OCH_3_), 103.21 (C-4), 109.62 (C-3), 111.09 (C-7), 112.58 (C-6), 124.87 (C-3a), 127.34 (C-o), 154.68 (C-5), 129.09 (C-m), 132.52 (C-7a), 132.86 (C-p), 134.39 (C-2), 139.16 (C-i), 145.35 (CH=N). HREIMS *m/z* [M + H]^+^ 344.106338 (calcd for C_17_H_17_N_3_O_3_S, [M + H]^+^ 344.10625).

### 4.5. In Vitro Anticancer Activity

#### 4.5.1. MTT Method

The cytostatic activity of the experimental compounds was investigated using an established methodology for assessing cell viability known as the Mosmann MTT method [45]. The assay is colorimetric and measures the activity of mitochondrial enzymes by reducing the yellow dye MTT (3-(4,5-dimethylthiazol-2-yl)-2,5-diphenyltetrazolium bromide) to violet formazan crystals. Exponential-phased cells were harvested and seeded (100 μL/well) in 96-well plates at 1.5 × 10^5^ density and incubated for 24 h. Cell cultures were treated and exposed to various concentrations (200–6.25 μM) of the tested compounds for 72 h, following which cell survival was quantified as percentage (%) relative to untreated control (100% cell viability).

#### 4.5.2. Statistical Methods

Experimental data were processed using nonlinear regression analysis in the GraphPad Prism^®^ software program. Semi-logarithmic “dose-response” curves were plotted and half-inhibitory concentrations (*IC*_50_) of the screened compounds were calculated for each of the tested tumor cell lines.

## Data Availability

Not applicable.

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
