# Peer review of "Novel Arylsulfonylhydrazones as Breast Anticancer Agents Discovered by Quantitative Structure-Activity Relationships"

_molecules, 2023, doi:10.3390/molecules28052058_

Round 1
Reviewer 1 Report
The current manuscript described arylsulfonyl hydrazones quantitative structure-activity relationship (QSAR) modeling analysis with the available activity data from previous reports and based on that data few arylsulfonylhydrazones 1a-1i were synthesized with a routine chemical procedure. The anticancer activities of the compounds were measured on human breast adenocarcinoma cell line MCF-7 and on the TNBC cell line MDA-MB-468. The reported compounds in silico studies were evaluated for physicochemical and ADME properties and pharmacokinetic (PK) parameters.
Presented studies in the manuscript and experimental data supporting the title and conclusion of the manuscript. However, according to the previous reports, the current manuscript has not improved comparatively and scientifically other than Insilco studies. Based on these the manuscript studies need major improvements and this reviewer recommends publishing in molecules after major revision or resubmission.
Recommendations
The author mentioned that based on QSAR Models they synthesized 1a -1i. Although the scope of substrates is very little, required to add more examples with different substitutions like fluoro, CF3, or H donors or acceptor groups on both sides of hydrazones. If possible do replace core indole with an Aza-indole kind of substituent
Also supposed to add activity profile against any specific protein target or kinases like pi3ks of breast cancers
Remove the word ‘Novel’ from the title, also authors unnecessarily repeat the words ‘Novel’ and ‘newly designed’ throughout the manuscript, which must be removed and rewritten in the required sentences. The sulfonylhydrazones were well-known compounds every time mentioning those words not meaning full.
Reviewer 2 Report
1- Line 17, sulfonylhydrazone is a one word while you write it phenyl hydrazone so please correct it allover the manuscript.
2- Line 22, write the exact number of the four compounds and their IC50 values.
3-Add the r2 values for your QSAR models in the abstract.
4- The introduction need one paragraph about QSAR types, significance,....etc.
5- Line 40 to 48, is a very long sentence, avoid such long sentences allover the manuscript.
6- Line 52-56, is a paragraph contain only 2 sentences so join these to the above paragraph.
7- Title of figure 1 need rephrasing and avoid repetition of information in the text above.
8- Line 106, you have to give reference to the program used for QSAR model.
9- Line 149, do not use 2 ands in one sentence.
10- Line 145, it is hydrazones not hydrazides
11- In the experimental part the 1HNMR assignments for compounds 1f and 1g are incorrect and are not compatible with the spectra in the supporting information you have to revise and rewrite.
12- In figure 2 add numbers for each compound inside the boiled egg.
Reviewer 3 Report
Novel Arylsulfonylhydrazones as Breast Anticancer Agents Discovered by Quantitative Structure-Activity Relationships
In this article, the authors present the synthesis and research on the anticancer activity of nine sulfonyl hydrazones. The newly synthesized compounds were designed based on QSAR models. It is a computational modeling method for revealing relationships between structural properties of chemical compounds and biological activities. The obtained results are interesting and may interest many readers. Although this method is of fundamental importance from the point of view of the presented results, there is practically no information about it in the Introduction. It seems, therefore, that it would be advisable to supplement the Introduction with information on this method.
The text shows a lack of consistency in the nomenclature of the synthesized compounds. In the title and most of the text, the authors use nomenclature derived from hydrazone derivatives, while in the Materials and Methods section, the names of synthesized compounds appear as hydrazides. From the point of view of nomenclature, the compounds obtained by the authors are actually hydrazide-hydrazones. Its name unsurprisingly comes from the dual possibility of reading the order of atoms in a moiety. If the nomenclature begins with the carbonyl group, we can say that we are dealing with monosubstituted carbonyl hydrazides, while if we regard the imine bond as the beginning, the-monosubstituted-hydrazone clearly appears in front of us. Therefore, authors should decide what form of naming they use and be consistent in it.
The authors should also supplement the list of cited literature with DOI numbers.
Also, there are some minor typos in the text that should be corrected:
· line 137 - the name Table S1 appears, but there is no such table in the text; it is included in the supplementary materials (as Table 1 and not Table S1) and information about it should be placed here.
· lines 365 and 421 - designation of the type of NMR spectrum 1H or 13C should contain digits as a superscript.
· line 432 - Celsius degree designation is incorrect.
Round 2
Reviewer 1 Report
In the revised version of the manuscript, the authors tried to address the major issues. However, the authors responded to the issues very superficially and did minor modifications to the manuscript that did not much improve it.
The authors need to make major changes as per the points raised previously.
